# Antibiotic Prescribing Patterns for Outpatient Pediatrics at a Private Hospital in Abu Dhabi: A Clinical Audit Study

**DOI:** 10.3390/antibiotics11121676

**Published:** 2022-11-22

**Authors:** Faris El-Dahiyat, Dalal Salah, Meriam Alomari, Abdullah Elrefae, Ammar Abdulrahman Jairoun

**Affiliations:** 1Clinical Pharmacy Program, College of Pharmacy, Al Ain University, Al Ain P.O. Box 64141, United Arab Emirates; 2AAU Health and Biomedical Research Center, Al Ain University, Abu Dhabi P.O. Box 112612, United Arab Emirates; 3The Life Corner, Abu Dhabi P.O. Box 35566, United Arab Emirates; 4Sheikh Shakhbout Medical City, Abu Dhabi P.O. Box 11001, United Arab Emirates; 5Colchester General Hospital, Essex, Colchester CO4 5JL, UK; 6Health and Safety Department, Dubai Municipality, Dubai P.O. Box 67, United Arab Emirates; 7School of Pharmaceutical Sciences, Universiti Sains Malaysia, Gelugor 11700, Malaysia

**Keywords:** antibiotics, pediatrics, prescribing pattern, drug utilization, WHO indicator

## Abstract

Background: Antibiotics are commonly used in pediatrics. The aim and objectives were to evaluate the antibiotic prescribing patterns of pediatric outpatients at a private hospital in Abu Dhabi, UAE. Methods: A retrospective drug utilization review was conducted for pediatric patients aged 1–18 between June and December 2018. The prescriptions with inclusion criteria were reviewed and evaluated by using the WHO indicators. Results: 419 encounters included were female (50.1%). Most pediatrics were aged 4–6 years (35.3%). The average number of drugs per prescription were 4.9 drugs. The percentage of parenteral medication prescriptions was 16.9%, and with antibiotic prescriptions was 43.0%, where cefaclor was the most prescribed antibiotic (31.1%). The average consultation time was 14 min, while the average dispensing time was 9.6 min. The most common diagnosis where antibiotics were prescribed was acute pharyngitis (33.4%). There were about 60.6% with lab investigation. Conclusion: As per the WHO indicators, the pediatric outpatient department has a high rate of antibiotic use and polypharmacy, but adherence to the drug formulary and prescribing medicines using generic names was appropriate. The average time for consultation and dispensing were suitable. Irrational antibiotic use for inappropriate diagnoses such as acute otitis media and bronchiolitis were found.

## 1. Introduction

The utilization of drugs plays an important role in helping the healthcare system to interpret, understand and improve the prescription, administration, and use of medication [1]. Irrational use of medications leads to an increased cost of treatment, adverse drug reactions, morbidity, and mortality. The WHO estimates that at least 50% of all medicines are used irrationally [1]. Irrational medication use can be in the form of self-medication, misuse of medications, polypharmacy, excessive use of antibiotics, and prescribing medication without following clinical guidelines. Our study evaluates the pediatric prescription patterns in the Abu Dhabi region, and the focus of this study is on antibiotic prescribing due to the concerns of newly introduced antibiotic stewardship and increasing antibiotic resistance as a global issue affecting both developing and developed countries, such as Europe, where high levels of resistance are associated with a high mortality rate [2]. High rates of antimicrobial use, where 50% and 85% of pediatric patients received antibiotics in developed and developing countries, respectively [3]. The current study aimed to assess the drug usage problem in a private health care facility (pediatric sectors) using selected WHO indicators and investigate the laboratory tests requested of pediatric patients for diagnosis because antibiotic choice should be based on investigative tests [4]. Different indicators were created, identified, standardized, and assessed by the World Health Organization (WHO) [5]. These are divided into three classes: prescribing indicators, patient care, and facility indicators [5,6]. Antibiotics illustrate the most widely used and prescribed therapeutic agents. The predominant drug prescription varies across ages; the most exposed age to the antibiotic are the preschool children [7] where physicians frequently prescribe more antibiotics than what is recommended and advised by guidelines for many causes: a variety of market announcements, inopportune use (such as the use for viral infections), pressure from patients to prescribe antibiotics and limited access to infectious disease specialists in outpatient departments [8]. The significant number of antibiotic prescriptions has possibly favored the growth and spread of antibiotic-resistant bacteria [9]. It has been shown that every year in the United States of America (USA), a large percentage of all antibiotics prescribed are given to children, and 50% of those may be unnecessary.

In comparison, in Canada, 74.0% of preschool children with respiratory infections have been prescribed an antibiotic where nearly 85.0% of these cases were prescribed inappropriately [9]. In the United Arab Emirates, UAE, it has been shown that healthcare is provided and attainable for all citizens all over the country through primary, secondary, and tertiary care. Dispensing drugs is regulated by the UAE health regulations, which prevents the dispensing drugs with incomplete or no prescriptions. There is a limited number of studies that review the prescribing patterns in the United Arab Emirates (UAE) [10,11], and none of these studies assess the prescribing patterns among pediatric patients in the UAE.

## 2. Materials and Methods

### 2.1. Study Design

A retrospective drug utilization review was carried out on a hospital between June 2018 and December 2018 in an outpatient pediatric private hospital in Abu Dhabi, UAE, to evaluate the implementation and compliance with WHO indicators for drug use in the health facilities. The hospital is one of the most extensive multispecialty medical facilities in the Abu Dhabi region. It acts as a teaching hospital for students majoring in health sciences while providing medical training to other personnel.

### 2.2. Target Population (Inclusion Criteria and Exclusion Criteria)

Pediatric patients of both genders, ages up to 18 years, attended the hospital between June 2018 and December 2018. All pediatric patients attended in the last six months of 2018 with at least one prescribed antibiotic.

#### 2.2.1. Inclusion Criteria

Pediatric patients aged less than 18 years. Patients received oral or parenteral antibiotics with or without other medications.

#### 2.2.2. Exclusion Criteria

The patients who received any topical prescription such as skin cream or skin ointments and eye drops, or eye ointments were excluded. Patients aged more than 18 years.

### 2.3. Sample Size

A study of a single facility should measure the exact percentage indicators with 95.0% confidence intervals of plus or minus 10.0% according to the number of patients that visited the hospital last year according to inclusion and exclusion criteria [12].

The number of pediatric patients who visited the hospital in the study period (June 2018 to December 2018) was 4740. After applying the inclusion and exclusion criteria, 2527 eligible patients were selected. 355 was the minimum required representative sample size using an online sample size calculator (Raosoft) with a confidence level of 95.0% and an acceptable margin of error of 5.0%. Using convenience sampling, 419 patient records were selected by choosing only the odd numbers of patients (Figure 1). However, the sample size used in this research was 419 patients, which exceeded the minimum representative sample with 95.0% confidence intervals of plus or minus 5.0%.

### 2.4. Data Collection Questionnaire and Procedure

The information was acquired from the electronic medical files of pediatric patients registered at the private hospital in Abu Dhabi, United Arab Emirates, for the outpatient clinic. We used a pre-made questionnaire modified from prior studies, a structured questionnaire created by the WHO, and common markers of antibiotic prescribing.

Demographic information, such as facility prescribing, and patient care indicators, which were the prescribing indicators for antibiotics, the most frequently prescribed antibiotics, and the prescribing factors for antibiotic prescription, were all included in the data collected to properly evaluate the prescription pattern. Rapid antigen testing, polymerase chain reaction (PCR), nucleic acid amplification testing, serological testing, and cultures were among the microbiological laboratory procedures that were examined to detect and identify respiratory pathogens [13,14].

During the study, the patients’ medical records were examined and any modifications to the drug chart or laboratory information were documented. The data collection procedure was covered in training, and two nurses and three clinical pharmacists were already familiar with the data contained in the data abstraction format. In addition, two data collection supervisors received training on the methodology for monitoring data collectors. Data spanning 419 medical records were gathered using a standardized data collection technology that had been tested beforehand. In this instance, data gatherers employed a special medical card identifying number to distinguish themselves and gather information from the chosen medical records.

### 2.5. Data Quality Control

Five percent of the sample was pre-tested before the final data collection process, and adjustments were made in response to the feedback and observations acquired. The data gatherers underwent daily intensive observation and training. After each data collection day, the accuracy of the collected data was ensured by reviewing the accuracy of the filled-out information. Any errors were rectified immediately when discovered.

### 2.6. Study Variables

The study’s dependent variable was the antibiotic prescribing pattern (WHO indicators). The independent variables were the sociodemographic factors (gender, age, and place of residence) and antibiotic knowledge (indication, frequency, and regimen/combination).

### 2.7. Operational Definitions and WHO Prescribing Indicators

Prescribing indicators:

a. The average number of drugs per case was calculated by dividing the total number of different drug products prescribed by the number of cases obtained.

b. The percentage of drugs prescribed by the generic name was determined by dividing the number of drugs prescribed by the generic name by the total number of drugs prescribed, multiplied by 100.

c. The percentage of cases where an antibiotic was prescribed.

d. The percentage of cases where an antibiotic was injection was calculated by dividing the number of patients cases multiplied by 100.

e. The percentage of drugs prescribed from the essential drug list (EDL) was determined by dividing the number of products prescribed from the EDL of the hospital by the total number of drugs prescribed, multiplied by 100.

2.Patient care indicators:

a. Average consultation time was determined by dividing the total time for a series of consultations by the actual number of consultations.

b. Average dispensing time was calculated by dividing the total time for dispensing drugs to a series of patients by the number of cases.

c. The percentage of drugs dispensed was calculated by dividing the number of drugs dispensed at the health facility by the total number of drugs prescribed, multiplied by 100.

3.Facility indicators:

a. Availability of a copy of the EDL or formulary: by stating yes or no.

b. Availability of drugs was calculated by dividing the number of specified products in stock by the total number of drugs on the checklist of essential drugs, multiplied by 100.

### 2.8. Ethical Consideration

Ethical approval and a permission letter to conduct the research were obtained from the health department of the private hospital.

### 2.9. Statistical Analysis

Statistical Package for Social Science (SPSS) version 20 was used for data analysis. Qualitative variables were reported using frequency counts and percentages, and quantitative data were calculated using descriptive statistics and presented as mean, median, and mode or according to the type of distribution of each variable.

## 3. Results

### 3.1. Demographic Characteristics

The population considered for this research study comprised 419 patients. Of the 419 patients; 210 (50.1%) were female, and 209 (49.9%) were male. Data of the patients showed that the age ranged from 1–18 years where most of the patients 148 (35.3%) were aged 4–6 years, followed by 137 (32.7%) patients aged 1–3 years, about 64 (15.3%) patients were aged 7–9 years, 46 (10.9%) were aged 10–12 years, 17 (4.1%) were aged 13–15 and only 7 (1.7%) were aged 16–18 years (Table 1).

### 3.2. Prescribing Indicators and the Number of Drugs Prescribed per Case

As a result of this study, the number of prescriptions studied was 419, with a total of about 2074 drugs prescribed and an average of 4.9 drugs per prescription. The percentage of cases where parenteral antibiotics were prescribed was 16.9%, where 71 prescriptions out of the 419 were injected. All prescribed medications were from a formulary found in the electronic system used in the hospital. The number of drugs prescribed per encounter varied from 1 drug per prescription to 13 drugs per prescription, about 9.5% for one drug prescribed, 7.9% for two drugs, 11.5% for three drugs, 12.4% for four drugs and about 58.7% for more than four drugs (Figure 2).

### 3.3. Percentage of Cases with Parenteral Antibiotics

The percentage of cases with parenteral antibiotics was 71 (16.9%), all by ceftriaxone injection. The other 348 (83.1%) case were administered without injection (Table 2).

### 3.4. Percentage of Cases with Antibiotic Prescirption

The percentage of cases with antibiotic prescription was 43.0%, where the most prescribed was cefaclor, 130 (31.1%), followed by co-amoxiclav, 103 (24.6%), then 71 (16.9%) ceftriaxone injection, and 69 (16.5%) azithromycin, these were the most prescribed antibiotics in this study (Table 3).

### 3.5. Patient Care Indicators, Average Consulting Time, and Average Dispensing Time

The average consultation time was 17 min, while the average dispensing time was 9.6 min. About 99.0% of the prescribed medications were dispensed. The knowledge of the parents of the correct dosage and their education were not assessed in this study because this study was a retrospective, not prospective study. The average consulting time was 17 min, ranging from 2 to 58 min, where 5.5% were seen in 18 min, and 5.5% in both 17 and 14 min. The mean was 23 min; the mode was 18 min, and the median was 18 min (Figure 3).

The average dispensing time was 9.6 min, with 19.8% in 5 min, 18.1% in 10 min and 16.2% in 12 min, 0.5% in 23 min, and 0.2% in 23 min. The average dispensing time, with 40.6%, was 6–10 min. This variation may differ according to the system and medication approval from the different insurance companies. The mean was 9.3, mode 5, and median 10 (Figure 4).

### 3.6. Facility Indicators

The hospital had an electronic system that depended on an electronic drug formulary and the British National Formulary (BNF). All prescriptions were electronic, containing information of the patients’ name, gender, age, history of allergy, and diagnosis.

### 3.7. Antibiotics Prescribing Patterns

The antibiotic prescribing pattern was also investigated in this study. The percentage of cases with antibiotic prescription was 43%, the percentage of antibiotics prescribed by their generic name was 100%, and the percentage of antibiotics from the formulary was 100%.

### 3.8. Distribution of Diagnoses for Which Antibiotics Were Prescribed

There were about twenty-six diagnoses used in this study. The most common diagnoses for which antibiotics were prescribed were acute pharyngitis (33.4%), acute tonsillitis (16.0%), otitis media (15.8%), acute bronchitis (6.7%) and bronchiolitis (6.2%) (Figure 5).

### 3.9. Prescribing of Antibiotics Based on Laboratory Tests

There were about 39.4% of lab tests requested with antibiotic prescription, such as rapid antigen tests, polymerase chain reaction (PCR), nucleic acid amplification tests, serological tests, and cultures, whereas about 60.6% of cases prescribed antibiotics without any lab tests (Figure 6).

## 4. Discussion

Four hundred and nineteen patients from the pediatric department in a private hospital in Abu Dhabi were reviewed over six months in this study. Out of the 419 patient prescriptions, 210 were female and 209 patients were male, almost the same percentage for both, a slight increase in female, similar to some studies [15,16], but disagreeing with other studies which reported higher percentages for males than females [17,18,19]. The most common age that were prescribed antibiotics in this study was between 4–6 years, followed by 1–3 years. This differs from other studies where the ages ranging from 1 month to 13 years were more prescribed [6,15,17,20]. It seems that most of the patients were of an age where they would be entering school, which is related to an increase in infections.

Regarding the prescribing indicators in this study, it was found that the average number of drugs prescribed per case was 4.9. This rate was higher than the WHO standard of 2.0 and similar to two studies in India where the average were 6.12 [18], and 4.2 [21]. This means that there was polypharmacy, which can lead to an increased risk of adverse drug interactions, antimicrobial resistance, dispensing errors, and non-adherence to medications. However, because of the availability of electronic systems, the risk of interaction was reduced as the electronic system can make an alert when there is any possibility of interaction. In addition, this seems to be a very-high rate compared to other studies. The average was higher than the WHO but less than our study as the study average in Nigeria was equal to 3.1 [15], in India 3.3 [15], in Yemen 3.2 [22], in Jordan 2.4 [23], Umm Al Quwain 2.6 [24], and in both Dubai and Saudi Arabia [25] more than 2, while a study conducted in Nepal [26] showed that average only 1.9 which is well within the range of the WHO. Another factor that makes our results differ from those of other studies, such as in Umm Al Quwain, Dubai, and Saudi Arabia, is that all these studied hospitals were government-run, while our hospital was privately owed. Hence, there is more restriction in the government sector hospitals.

Continuing the assessment of the prescribing indicators, it was found that the percentage of cases where antibiotics were prescribed was 43.0%, higher than recommended by the WHO for antibiotic prescriptions which should be less than or equal to 30.0% of the total number of patient. This study showed that there was an increase in using antibiotics in pediatric patients. Such a use of antibiotic is similar in Umm Al Quwain [24], India [17], Zambia [27], Nigeria [15], Yemen [22], Jordan [23], Saudi Arabia [25], India [28], Italy, the UK and the Netherlands [29], as well as higher than in Saudi Arabia [30], Georgetown [19], India [18], and the USA [20].

On the other hand, the most prescribed antibiotics in this study were Cefaclor 130 (31.1%), Co-amoxiclav 103 (24.6%), and ceftriaxone 69 (16.9%), while the less commonly prescribed were amoxicillin 1 (0.2%) and clarithromycin 2 (0.5%). When comparing these results with other studies, it was found that some studies have shown that the most commonly prescribed antibiotic wee cephalosporins [18], penicillin [27], macrolides [31], azithromycin, amoxicillin–clavulanic [23], and azithromycin [32]. Most studies showed that amoxicillin was the most prescribed antibiotic and that there is an increasing use of broad-spectrum beta-lactams [33]. Another study found that amoxicillin, azithromycin, Amoxicillin–clavulanic, and cefaclor (11.7%) were most prescribed [16]. This is similar to our study regarding the type of antibiotics. In another study, amoxicillin prescription was also high in both the UK and the Netherlands [29]. Two studies showed a high prescription of amoxicillin [6,19]; thus, all these studies were different from our study where amoxicillin was almost never prescribed. Compared to other studies we found a marked preference for prescribing cefaclor in this study and the reasons for this are unclear. They may be related to differences in patient characteristics, prescribing practices by doctors or the cost of medication, in antibiotic prescribing guidelines for most of the differential diagnosis, such as acute sinusitis, acute otitis media, and pharyngitis amoxicillin is the first-line treatment unless the patient has an allergy. However, in this study, there was no amoxicillin use.

The percentage of cases with parenteral antibiotic use was high in this study 16.9%, which is above the 10.0% WHO standards. The results in this study were similar to studies in India [18,34], Sierra Leone [1], and Ethiopia [35], but more than studies conducted where it was less than 10.0% [15,17,25]. Overusing parenteral antibiotics may lead to increasing costs and the transmission of infections because of the possibility of needle stick injuries. Due to this, it is better to choose an oral form of administration to reduce costs and the possibility of infections if applicable [5]. About 100% of the drugs were prescribed from a formulary that was available in the system, encouraging doctors to select medications from the system, helping to select appropriate drugs according to the prevalence of disease. Adhering to a drug formulary can help in rational prescribing, similar observations were seen in another study [24] but different from other studies in India [18], Nigeria [15], and Georgetown [19].

About 100% (419) of the antibiotics were prescribed by their generic names, where the use of generic names is recommended to decrease the cost of medication to the patient. As generic prescribing is mandatory by the health authority as per the circular U.S./27/18, published on 23 July 2018, for regulating the dispensing of generic medicine, the DOH encourages healthcare facilities to start offering more generic medicine options to provide patients with better value for money. Many countries worldwide have successfully encouraged the use of generic medicine over branded medicine. In Germany, for example, 80.0% of prescribed drugs are generic, whereas in the UK 78.0% of prescribed drugs are generic [36] and in comparison to the results in this study in the UAE they follow the DOH in using generic names over the brand names.

The average consultation time in our study was 17 min, considered suitable as per the WHO standards, and longer consultation time significantly improved patient satisfaction, effective communication, and practical resource use. This result seems to be similar to study [24] with an average consultation time of 20 min, which differs from studies conducted with an average consultation time less than 10 min, which is the WHO standard [21,34,35]. The average dispensing time was found to be 9.6 min and considered to be suitable because a short dispensing time will affect appropriate labeling, the information given to parents regarding medications, and appropriate education, the results of this study were similar to other studies where average dispensing times were 12 min, 9–12 min [24,34] and 7 min [21] while differing from results of other studies where the average dispensing time was only 1 min [35].

The most common diagnoses for which drugs were prescribed were acute pharyngitis (33.4%), acute tonsillitis (16.0%), otitis media (15.8%), acute bronchitis (6.7%) and bronchiolitis (6.2%), this means most of the diagnoses were for respiratory tract infections, similar to most of the studies conducted in different countries all over the world [17,18,27,31]. Additionally, a study found that the most common diagnosis was respiratory infections with bronchitis, and in another study found that tonsillitis as the most common [16]. Another study mentioned that upper respiratory tract infections (URTI), urinary tract infections (UTI), and gastroenteritis were the top diagnoses [23]. In Australia, above 50.0% of antibiotic prescriptions were contrary to the guidelines. Patients commonly presented with acute otitis media, sore throat, acute exacerbation of asthma, and a cough [37]. There were some cases of diagnosis with antibiotic treatment relating to viral infections and thus no need to prescribe an antibiotic. This is considered to be an irrational use of an antibiotic that can lead to antibiotic resistance. Empiric antibiotic therapy in these cases is unnecessary. This inappropriate use can lead to antimicrobial resistance among bacterial pathogens encountered in pediatrics. Another important issue is that one study documented that the physicians felt pressured by parents to prescribe antimicrobials for respiratory infections [38].

In our study 60.6% of the cases where antibiotics was prescribed ordered lab investigation for pediatric patients, such as rapid antigen tests, polymerase chain reaction (PCR), nucleic acid amplification tests, serological tests, and cultures. Whereas about 39.4% of cases prescribed antibiotics without any lab investigation, indicating that many antibiotics were prescribed without any confirmation of bacterial infection, which can lead to increased antibiotic resistance. Lab investigation is crucial because it can facilitate the decision of antibiotic use, whether to be de-escalated, broadened, stopped, or identify viral etiology in some cases [39]. A study showed that the availability of an accurate and rapid diagnostic method for common respiratory viruses was associated with decreased use of antimicrobials in infants and children hospitalized for viral respiratory illnesses [40].

Introducing learning modules for physician in ambulatory care such as outpatient settings in the hospitals can decrease unnecessary antibiotic prescriptions for acute respiratory infections [41]. Furthermore, implementing pediatric common infection pathways by physicians will reduce variability in infectious disease management which in turn will shorten antibiotic duration, unnecessary antibiotic use, and antimicrobial resistance in pediatrics [42]. Physicians in Saudi Arabia use local guidelines specific for antibiotic prescribing for community-acquired respiratory tract infections [43], which is a starting point in the region to develop local guidelines to manage infectious diseases appropriately.

## 5. Limitation of the Study

Our study has some limitations. First, the study was carried out in only one study site, which is a private hospital in Abu Dhabi; therefore, more studies are required from other different hospitals, either private or government, to evaluate different prescribing patterns with more findings. Secondly, the study was conducted in an outpatient setting during the end summer and the start of winter, so the pattern may change with seasonal variation and may differ in an inpatient setting. Thirdly, there was an over-representation of children aged 4 to 6, which may not be generalizable to the pediatric population. Because the study was a retrospective design, poor documentation and incomplete information should be considered. A further limitation of the current study is the convenience sample, which is linked to the difficulty of generalizing the survey results to a larger population and the potential for under- or over-representing the population.

## 6. Recommendation

Based on the finding of this study the following recommendations are made to address several issues and help in any further research. To encourage the regulatory authorities around the world to increase their restriction of polypharmacy, the mean number of drugs per prescription should be as low as possible since a higher number of drugs increase the risk of drug interaction, risk of bacterial resistance, non-compliance, and cost. Most patients should receive their medication for their empiric diagnosis and this prescribing pattern of dispensing should be convenient and depend on lab results for investigation. Rapid and accurate viral and bacterial diagnostic testing should be an important component of physicians’ plans for diagnosing any infection. To decrease the antibiotic use for infections patients should be educated about the dangers and limited benefits of their use, and clinicians should consider appropriate responses to these different patient pressures to prescribe antibiotics. Other studies are needed to evaluate physicians’ practices on how to use and follow guidelines. To promote the rational use of antibiotics we recommend to publish standardized guidelines establishing therapeutic committees, targeting continued medical education and drug-use evaluation at an institute level, in addition to policymakers to enhance prescription practices.

## 7. Conclusions

When comparing our results to the WHO standards of drug prescription patterns in the outpatient pediatric department in the hospital. There was a high rate of antibiotic and injection use. Additionally, there was the presence of polypharmacy compared with the standard rates, although it was lower than some other countries in the world. The strength of our study was the adherence to the drug formulary and prescribing of medicines using generic names. This will help in keeping the cost of treatment low. The average time for both consultation and dispensing was suitable, and this improves patient satisfaction. This adherence is because of enforcing a DOH to use both generic names and formularies.

## Figures and Tables

**Figure 1 antibiotics-11-01676-f001:**
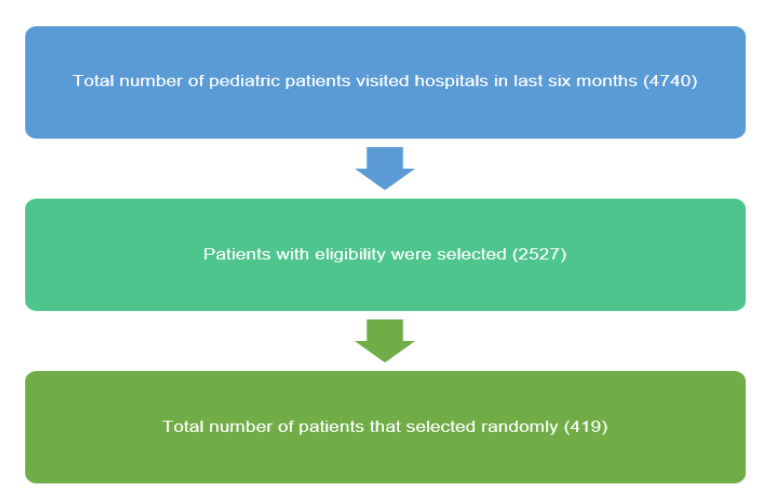
The sample size of pediatric patients.

**Figure 2 antibiotics-11-01676-f002:**
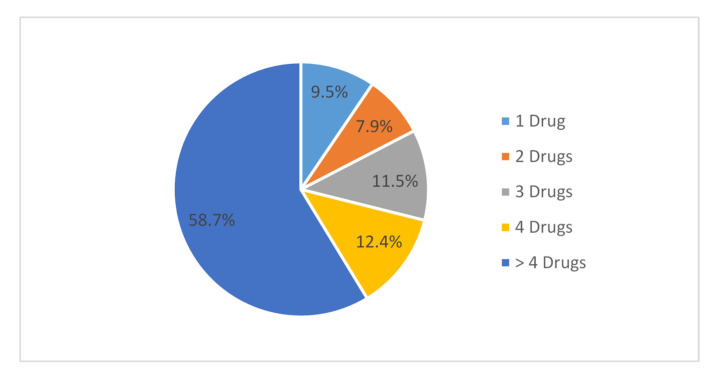
Percentage of the number of drugs prescribed per case.

**Figure 3 antibiotics-11-01676-f003:**
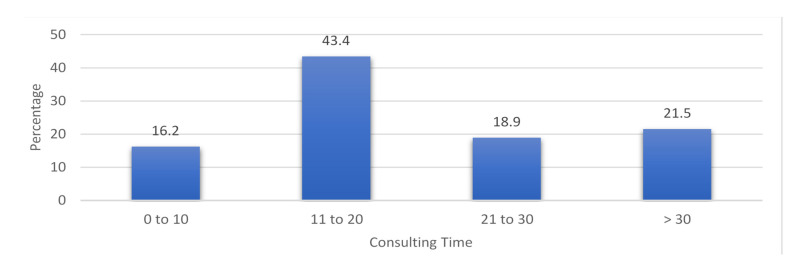
Percentage of consulting time.

**Figure 4 antibiotics-11-01676-f004:**
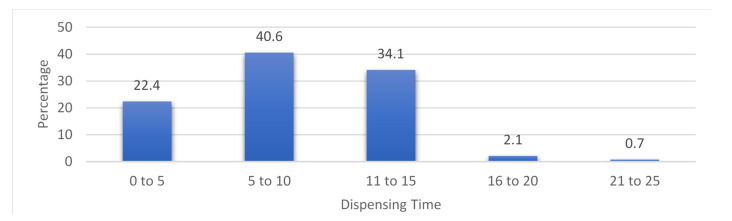
Percentage of dispensing time.

**Figure 5 antibiotics-11-01676-f005:**
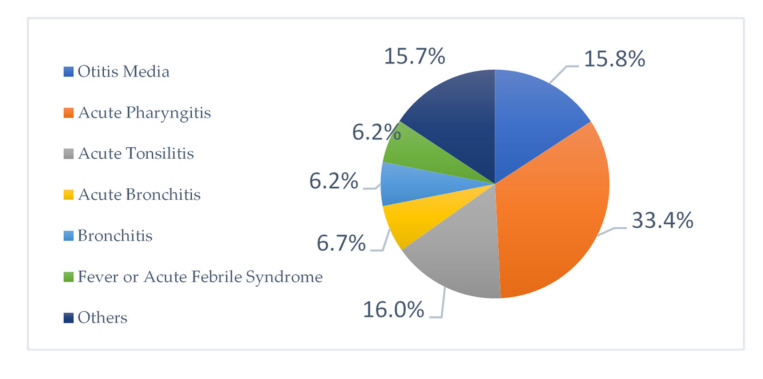
Distribution of diagnoses for which antibiotics were prescribed.

**Figure 6 antibiotics-11-01676-f006:**
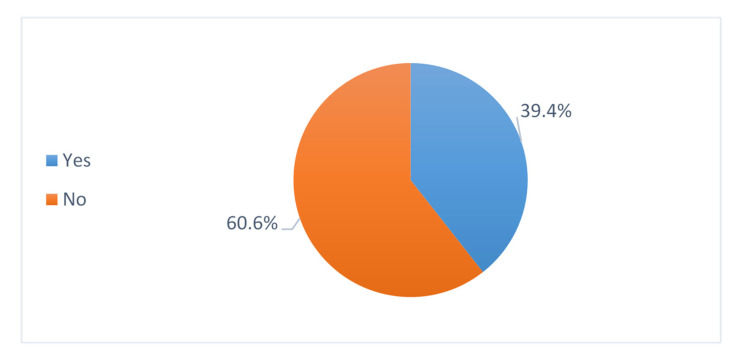
Prescribing of antibiotics based on laboratory tests.

**Table 1 antibiotics-11-01676-t001:** Frequency and percentage of age.

Age	Frequency	Percent
**1–3**	137	32.7
**4–6**	148	35.3
**7–9**	64	15.3
**10–12**	46	10.9
**13–15**	17	4.1
**16–18**	7	1.7
**Total**	419	100

**Table 2 antibiotics-11-01676-t002:** Percentage of cases with parenteral antibiotics.

Injections	Frequency	Percent
**No**	348	83.1
**Yes**	71	16.9
**Total**	419	100

**Table 3 antibiotics-11-01676-t003:** Frequency and percentage of cases with antibiotics.

Antibiotic	Frequency	Percent
**Amoxicillin**	1	0.2
**Azithromycin**	69	16.5
**Cefaclor**	130	31.1
**Cefdinir**	15	3.6
**Cefixime**	8	1.9
**Cefpodoxime**	9	2.1
**Ceftriaxone**	71	16.9
**Cefuroxime**	11	2.6
**Clarithromycin**	2	0.5
**Co-amoxiclav**	103	24.6
**Total**	419	100

## Data Availability

The data that support the findings of this study are available from the corresponding author upon reasonable request.

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
