# Peer review of "Antibiotic Prescribing Patterns for Outpatient Pediatrics at a Private Hospital in Abu Dhabi: A Clinical Audit Study"

_antibiotics, 2022, doi:10.3390/antibiotics11121676_

Round 1
Reviewer 1 Report
Thank you for the opportunity to review this manuscript.
In general, the subject tackled by the authors is of high importance, especially in pediatric outpatient antimicrobial stewardship which by itself presents a big challenge to achieve. It is difficult to measure and review the outcomes of outpatient protocols. Therefore, the authors raise a very important issue that I believe can be better presented to readers and more organized in a way to reach more audience.
The manuscript can shortened and it would be beneficial. the introduction should be better structured, that is similar ideas grouped in the same paragraphs, line 60-69 for example should be mentioned earlier in introduction or kept for discussion]/conclusion.
the methods can be also more concise to the point yet include all the terms the authors use, it is confusing as such. Moreover, how is it counted if an encounter received 2 anitbitoics ? if the same encounter was a follow up visit after a week let's say, how is that entered in the data? it would also be interesting to match the diagnosis with the age group and the ordered antibiotics. another interesting point would be to see if a longer consulting time led to fewer antibiotic prescription, that is, if the conslutant took their time in the consult, did this result in better stewardship?
I think these can be restructured and addressed in manuscript, seeing that DDD was not used as a metric, other metrics are albeit misleading or not very well known(dispensing time? what benefit does this have?)
the english needs improvement, many run-on sentences
Author Response
Reviewer 1
Comments and Suggestions for Authors
- Thank you for the opportunity to review this manuscript.
- In general, the subject tackled by the authors is of high importance, especially in pediatric outpatient antimicrobial stewardship which by itself presents a big challenge to achieve. It is difficult to measure and review the outcomes of outpatient protocols. Therefore, the authors raise a very important issue that I believe can be better presented to readers and more organized in a way to reach more audience.
Author comments: Thank you for these encouraging words – much appreciated
- The manuscript can shortened and it would be beneficial. the introduction should be better structured, that is similar ideas grouped in the same paragraphs, line 60-69 for example should be mentioned earlier in introduction or kept for discussion]/conclusion.
Author comments: Thank you – now addressed.
- The methods can be also more concise to the point yet include all the terms the authors use, it is confusing as such.
Author comments: Thank you – you are right. We have enhanced this section of the paper to discuss more details on data collection, its quality control, data reporting and analysis and hope this is now OK.
- Moreover, how is it counted if an encounter received 2 anitbitoics ? if the same encounter was a follow up visit after a week let's say, how is that entered in the data?
Author comments: thank you for this, in fact during data collection there was no more than one antibiotic prescribed per encounter and that based on Data Quality Control. Please refer to section 2.5 in method and material, and hope this is now OK.
- It would also be interesting to match the diagnosis with the age group and the ordered antibiotics.
- Another interesting point would be to see if a longer consulting time led to fewer antibiotic prescription, that is, if the consultant took their time in the consult, did this result in better stewardship?
- I think these can be restructured and addressed in manuscript, seeing that DDD was not used as a metric, other metrics are albeit misleading or not very well known(dispensing time? what benefit does this have?)
- The English needs improvement, many run-on sentences
Author comments: Thank you – now addressed, Proofreading and grammar check for the final version was done.
Reviewer 2 Report
This study is interesting in that it shows the use of antimicrobials for pediatric patients in other countries. However, several concerns can be considered.
The authors indicate the problem of polypharmacy, but due to the lack of information, it is impossible to determine if the drugs are unnecessary for the patients. The primary disease of these patients should be indicated and the details of non-antimicrobial prescriptions should also be presented and discussed.
The authors discuss the rate of antimicrobial use and the content of antimicrobials compared to WHO recommendations. However, the authors do not assess the appropriateness of antimicrobials in this population and therefore cannot evaluate their validity. Since the diagnostic names of infectious diseases are also investigated, the overall presentation of the results obtained should be re-examined.
The authors investigated the presence or absence of laboratory tests, but the appropriateness should be discussed with a detailed indication of what laboratory tests. (ex. Gram stain, blood culture, antigen testing for GAS, etc.)
In the study limitations, the authors state that they cannot generalize. The sampling methodology in this study may be problematic. Since the number of patients before sampling is large, a sensitivity analysis should be performed or the sampling method should be re-examined.
Author Response
Reviewer 2
Comments and Suggestions for Authors
- This study is interesting in that it shows the use of antimicrobials for pediatric patients in other countries. However, several concerns can be considered.
Author comments: Thank you for these encouraging words –we are pleased to answer your comments.
- The authors indicate the problem of polypharmacy, but due to the lack of information, it is impossible to determine if the drugs are unnecessary for the patients. The primary disease of these patients should be indicated and the details of non-antimicrobial prescriptions should also be presented and discussed.
Author comments: thank you for this, the main objective of this study is to evaluate the antibiotic prescribing pattern for pediatric patients. The polypharmacy and related outcomes is away form study focus and this could be considered in future studies and hope this is now OK.
- The authors discuss the rate of antimicrobial use and the content of antimicrobials compared to WHO recommendations. However, the authors do not assess the appropriateness of antimicrobials in this population and therefore cannot evaluate their validity. Since the diagnostic names of infectious diseases are also investigated, the overall presentation of the results obtained should be re-examined.
- The authors investigated the presence or absence of laboratory tests, but the appropriateness should be discussed with a detailed indication of what laboratory tests. (ex. Gram stain, blood culture, antigen testing for GAS, etc.)
Author comments: thank you. In fact, this study is a clinical Audit study aimed to assess and measure the compliance to the specified guidelines.
1-Therefore, current study measured how much the specific hospital is adhere to the WHO indicators for drug use in health facilities which is consistent with Many previous studies conducted similar methodology, the only difference is the study area.
2- The current study aimed to measure the magnitude of irrational antibiotic use in pediatric patents to fill gap between local regulation and practice and hope this is now acceptable.
Please find the previous similar studies:
- Sharma S, Bowman C, Alladin-Karan B, Singh N. Antibiotic prescribing patterns in the pediatric emergency department at Georgetown Public Hospital Corporation: a retrospective chart review. BMC infectious diseases. 2016 Dec;16(1):1-6.
- Yehualaw A, Taferre C, Bantie AT, Demsie DG. Appropriateness and Pattern of Antibiotic Prescription in Pediatric Patients at Adigart General Hospital, Tigray, Ethiopia. BioMed Research International. 2021 Apr 10;2021.
- Tadesse TY, Molla M, Yimer YS, Tarekegn BS, Kefale B. Evaluation of antibiotic prescribing patterns among inpatients using World Health Organization indicators: A cross-sectional study. SAGE open medicine. 2022 May;10:20503121221096608.
- Aldabagh A, Farha RA, Karout S, Itani R, Hammour KA, Alefishat E. Evaluation of Drug Use Pattern in Pediatric Outpatient Clinics in a Tertiary Teaching Hospital Using WHO Drug-Prescribing Indicators. Journal of Multidisciplinary Healthcare. 2022;15:1143.
- In the study limitations, the authors state that they cannot generalize. The sampling methodology in this study may be problematic. Since the number of patients before sampling is large, a sensitivity analysis should be performed or the sampling method should be re-examined.
Author comments: thank you, we agree with the reviewer. We have now updated the manuscript to address the use of convenience sampling as study limitation, its implications and the related strengths and weaknesses. We hope this is now OK.
Reviewer 3 Report
Line 41: the heaviness of patients, and inopportune use (like use for viral infections).
Pressure from patients and treating viral infections.
Results 3.3. Injections change to parenteral throughout.
Table 3. Gender, Antibiotic.
Table 5. Fever or acute febrile syndrome
3.9 Please describe laboratory tests, cultures performed, and rapid tests?
Discussion, Line 311. Describe the laboratory tests performed.
5. Limitations of the study, Line 326,
Because the study is a retrospective design, poor documentation and incomplete information considered to be considered.
Because the study is a retrospective design, poor documentation and incomplete information should be considered.
The following references should be reviewed and commented on both in the introduction and the discussion:
Antibiotic Stewardship
El Feghali RE. Infect Control Hosp Epidemiol 2021; 22:1.
Monsees EA. Am J Infect Control 2021; 49:1543.
Ward MJ. Appl Clin Inform 2021; 12:34.
Gerber JS. Pediatrics 2021; 147:e2020040295.
Horner C. JAC Antimicrob Resist 2021; 3:dlab029.
Fleming-Dutra KE. JAMA 2016; 315:1864.
Kronman MP. Pediatrics 2020; 146:e20200038.
Global Burden
Lancet 2022; 399:629.
Wang X. Lancet Glob Health 2020; 8:e497.
Resistance
Arnolda G. BMC Pediatr 2020; 20:185.
Ubukata K. J Med Microbiol 2020; 69:443.
Butler DF. Infect Dis Clin North Am 2018; 32:119.
Otitis Media
Hoberman A. N Engl J Med 2016; 375:2446.
Heikkinen T. Clin Microbiol Rev 2003; 16:230.
Vaccine
Leach AJ. BMC Pediatr 2021; 21:117.
Saudi Arabia
Torum Kuney D. J Antimicrob Chemother 2022; 77:i70.
Rapid Diagnostic Tests/Point of care (POC)
Cohen R.Pediatr Infect Dis J 2021; 40:674.
Mengelle C. J Clin Virol 2014; 61:411.
Author Response
Reviewer 3
Comments and Suggestions for Authors
- Line 41: the heaviness of patients, and inopportune use (like use for viral infections).
- Pressure from patients and treating viral infections.
- Results 3.3. Injections change to parenteral throughout.
- Table 3. Gender, Antibiotic.
- Table 5. Fever or acute febrile syndrome
- 9 Please describe laboratory tests, cultures performed, and rapid tests?
- Discussion, Line 311. Describe the laboratory tests performed.
- Limitations of the study, Line 326,
- Because the study is a retrospective design, poor documentation and incomplete information considered to be considered.
- Because the study is a retrospective design, poor documentation and incomplete information should be considered.
- The following references should be reviewed and commented on both in the introduction and the discussion:
Author comments: Thank you – now rephrased. We hope this is now OK.
Antibiotic Stewardship
- El Feghali RE. Infect Control Hosp Epidemiol 2021; 22:1.
- Monsees EA. Am J Infect Control 2021; 49:1543.
- Ward MJ. Appl Clin Inform 2021; 12:34.
- Gerber JS. Pediatrics 2021; 147:e2020040295.
- Horner C. JAC Antimicrob Resist 2021; 3:dlab029.
- Fleming-Dutra KE. JAMA 2016; 315:1864.
- Kronman MP. Pediatrics 2020; 146:e20200038.
Global Burden
- Lancet 2022; 399:629.
- Wang X. Lancet Glob Health 2020; 8:e497.
Resistance
- Arnolda G. BMC Pediatr 2020; 20:185.
- Ubukata K. J Med Microbiol 2020; 69:443.
- Butler DF. Infect Dis Clin North Am 2018; 32:119.
Otitis Media
- Hoberman A. N Engl J Med 2016; 375:2446.
- Heikkinen T. Clin Microbiol Rev 2003; 16:230.
Vaccine
- Leach AJ. BMC Pediatr 2021; 21:117.
Saudi Arabia
- Torum Kuney D. J Antimicrob Chemother 2022; 77:i70.
Rapid Diagnostic Tests/Point of care (POC)
- Cohen R.Pediatr Infect Dis J 2021; 40:674.
- Mengelle C. J Clin Virol 2014; 61:411.
Author comments: Thank you – now addressed and related references were cited, we hope this is now OK.
Reviewer 4 Report
The authors have provide an overview of outpatient antibiotic prescribing pattern for children in a private hospital in UAE.
The paper needs extensive revision for language and grammar
Abstract
Line 12: suggest adding UAE after Abu Dhabi
Round up all percentages to same decimal place
The phrase 'Encounters percentage with injections was (16.95%) ...' is ambiguous
Main text
Remove hyphens where not required (e.g. Line 33: anti-biotics should be antibiotics)
Line 84: preferable to replace '.. age from newborn to 18 years ..' with '... aged less than 18 years ..'
Calculation of sample size is ambiguous
An analysis of the 2527 patients that met the eligibility criteria, rather than a sample of 419 patients, would have strengthened the paper
Line 108: remove the word 'letter' to read: 'An ethical approval and permission to conduct ..'
Author Response
Reviewer 4
Comments and Suggestions for Authors
- The authors have provide an overview of outpatient antibiotic prescribing pattern for children in a private hospital in UAE.
Author comments: Thank you for these encouraging words –we are pleased to answer your comments.
- The paper needs extensive revision for language and grammar
Author comments: Thank you – now addressed, Proofreading and grammar check for the final version was done. Please see the proofreading certificate.
Abstract
- Line 12: suggest adding UAE after Abu Dhabi
- Round up all percentages to same decimal place
- The phrase 'Encounters percentage with injections was (16.95%) ...' is ambiguous
Main text
- Remove hyphens where not required (e.g. Line 33: anti-biotics should be antibiotics)
- Line 84: preferable to replace '.. age from newborn to 18 years ..' with '... aged less than 18 years ..'
- Calculation of sample size is ambiguous
- An analysis of the 2527 patients that met the eligibility criteria, rather than a sample of 419 patients, would have strengthened the paper
- Line 108: remove the word 'letter' to read: 'An ethical approval and permission to conduct ..'
Author comments: Thank you – now reworded. We hope this is now OK.
Reviewer 5 Report
In this manuscript, Faris EI-Dahiyat et al evaluated the antibiotic prescribing pattern outpatient pediatrics at a private hospital in Abu Dhabi. With the retrospective drug utilization review conducted for pediatric age(1-18) years between June and December 2018, the authors found the pediatric outpatient department has ahigh rate of antibiotic use and polypharmacy, but with adherence to drug formulary and prescribing of medicine using generic names. They also found time for consultation and dispensing were suitable. However, irrational use of antibiotics of inappropriate diagnoses such as acute otitis media and bronchiolitis exist. In this study, the authors described the information of samples, including the number, gender, as well as age distribution. They also mentioned the distribution of drug numbers per prescription, percentage with injections, antibiotic, and frequency of antibiotics. The consultation and dispensing times were analyzed in this study. Moreover, the authors analyzed frequency of disease in antibiotic use and percentage of lab investigation in antibiotic diagnosis, which indicated improper use of antibiotic without accurate confirmation of bacterial infection. Importantly, the authors also discussed on limitations for this study, which include one study site of private hospital, study conducted only in outpatient setting during summer time, overrepresentation of children aged 4 to 6 years old, and poor documentation and incomplete information in this study.
This study is very important for understanding antibiotic prescribing pattern and provide recommendations for rational antibiotic use.
Specific comments:
This study analyzed antibiotic prescribing pattern for outpatient’s pediatrics at private hospital in Abu Dhabi, the data was properly presented, results and conclusions are clearly discussed. However, the language in the manuscript needs improved, and easy to make readers confused, such as in line 264, 267..
Author Response
Reviewer 4
Comments and Suggestions for Authors
- In this manuscript, Faris EI-Dahiyat et al evaluated the antibiotic prescribing pattern outpatient pediatrics at a private hospital in Abu Dhabi. With the retrospective drug utilization review conducted for pediatric age(1-18) years between June and December 2018, the authors found the pediatric outpatient department has ahigh rate of antibiotic use and polypharmacy, but with adherence to drug formulary and prescribing of medicine using generic names. They also found time for consultation and dispensing were suitable. However, irrational use of antibiotics of inappropriate diagnoses such as acute otitis media and bronchiolitis exist. In this study, the authors described the information of samples, including the number, gender, as well as age distribution. They also mentioned the distribution of drug numbers per prescription, percentage with injections, antibiotic, and frequency of antibiotics. The consultation and dispensing times were analyzed in this study. Moreover, the authors analyzed frequency of disease in antibiotic use and percentage of lab investigation in antibiotic diagnosis, which indicated improper use of antibiotic without accurate confirmation of bacterial infection. Importantly, the authors also discussed on limitations for this study, which include one study site of private hospital, study conducted only in outpatient setting during summer time, overrepresentation of children aged 4 to 6 years old, and poor documentation and incomplete information in this study.
- This study is very important for understanding antibiotic prescribing pattern and provide recommendations for rational antibiotic use.
- Specific comments:
- This study analyzed antibiotic prescribing pattern for outpatient’s pediatrics at private hospital in Abu Dhabi, the data was properly presented, results and conclusions are clearly discussed. However, the language in the manuscript needs improved, and easy to make readers confused, such as in line 264, 267..
Author comments: Thank you for these encouraging words – much appreciated
Round 2
Reviewer 1 Report
Thank you for addressing the changes, the quality has approved, what remains is corrections related to the English language such as run-on sentences, grammatical and so on.
Author Response
Comments and Suggestions for Authors
Thank you for addressing the changes, the quality has approved, what remains is corrections related to the English language such as run-on sentences, grammatical and so on.
Author Response: Thank you for these encouraging words, proofreading and grammar check for the final version was done.
Reviewer 2 Report
I feel disappointed that I did not add more data to my comments, but the authors have generally responded to the points made.
Author Response
Comments and Suggestions for Authors
I feel disappointed that I did not add more data to my comments, but the authors have generally responded to the points made.
Author comments: Thank you for these encouraging words – much appreciated!
Reviewer 3 Report
Sentence lines 49 to line 53 should be restructured, eliminate heaviness, and look for a better definition of problematic patients; in the second portion of the sentence, some items are repeated.
Author Response
Comments and Suggestions for Authors
Sentence lines 49 to line 53 should be restructured, eliminate heaviness, and look for a better definition of problematic patients; in the second portion of the sentence, some items are repeated.
Author comments: Tank you – now rephrased
Reviewer 4 Report
The authors have not completely addressed the queries raised in the initial review.
- Suggest that the authors round up all percentages to same decimal place in the abstract and main text (e.g. 50.1% or 50%; 16.9% or 17% etc)
- In the abstract section: 'Encounters percentage with parenteral was (17 %) and the encounters parentage 16 with antibiotics was (43 %) where Cefaclor was the most prescribed (31 %)' seems ambiguous and should be rewritten.
Author Response
Comments and Suggestions for Authors
The authors have not completely addressed the queries raised in the initial review.
- Suggest that the authors round up all percentages to same decimal place in the abstract and main text (e.g. 50.1% or 50%; 16.9% or 17% etc)
- In the abstract section: 'Encounters percentage with parenteral was (17 %) and the encounters parentage 16 with antibiotics was (43 %) where Cefaclor was the most prescribed (31 %)' seems ambiguous and should be rewritten.
Author comments: Thank you – we have now updated the manuscript to address inconsistencies and hope this is now acceptable.